# Prevalence and social determinants of anxiety and depression among adults in Ghana: a systematic review and meta-analysis protocol

Victoria Awortwe [iD],[1] Meena Daivadanam [iD],[2] Samuel Adjorlolo [iD],[3] Erik MG Olsson [iD],[4] Chelsea Coumoundouros [iD],[1] Joanne Woodford [iD][1]

[1]Healthcare Sciences and e-Health, Department of Women's and Children's Health, Uppsala University, Uppsala, Sweden
[2]Global Health and Migration Unit, Department of Women's and Children's Health, Uppsala University, Uppsala, Sweden
[3]Department of Mental Health, School of Nursing and Midwifery, College of Health Sciences, University of Ghana, Accra, Ghana
[4]Cardiovascular Psychology, Department of Women's and Children's Health, Uppsala University, Uppsala, Sweden

**Correspondence to**
Victoria Awortwe;
victoria.awortwe@uu.se

## ABSTRACT

**Introduction** Anxiety and depression pose a significant global health challenge, especially affecting adults in low-income and middle-income countries. In many low-income and middle-income countries, including those in sub-Saharan Africa, social determinants such as access to affordable health services, conflict, food insecurity, and poverty may be associated with the prevalence of anxiety and depression, further contributing to health disparities. To mitigate the burden of anxiety and depression in sub-Saharan Africa, it is essential to develop country-level tailored mental health policies and strategies. For example, Ghana is working towards improving mental health via its 12 year Mental Health policy launched in 2021. However, the prevalence of anxiety and depression among adults in Ghana, along with associated social determinants remains largely unknown, posing challenges for mental health planning, resource allocation and developing targeted interventions. This systematic review seeks to (1) examine the prevalence of anxiety and depression among adults in Ghana and (2) explore social determinants potentially associated with anxiety and depression.

**Methods and analysis** Electronic databases (eg, African Index Medicus, CINAHL, EMBASE, MEDLINE, and PsycINFO) will be searched with all screening steps conducted by two independent reviewers. Secondary search strategies, including grey literature searches, will be used. Studies reporting on the prevalence of anxiety, depression and/or a combined symptom measure (ie, psychological distress) among adults in Ghana, using validated instruments will be included. If data allows, random-effects-meta-analyses will be performed to estimate pooled prevalence rates of anxiety and depression. Potential clinical and methodological moderators will be examined using subgroup analyses and meta-regression. A narrative synthesis will explore social determinants potentially associated with anxiety and depression among adults in Ghana.

**Ethics and dissemination** Ethical approval is not required as no primary data will be collected. Results will be disseminated via a peer-reviewed publication and presentations at academic conferences. Plain language summaries will be provided to relevant non-governmental organisations working in Ghana.

**PROSPERO registration number** CRD42023463078

## STRENGTHS AND LIMITATIONS OF THIS STUDY

⇒ This review, to the best of our knowledge, is the first of its kind, aiming to examine the prevalence of anxiety and depression among adults in Ghana and explore the social determinants potentially associated with anxiety and depression.

⇒ The review protocol adheres to quality standards informed by the Preferred Reporting Items for Systematic reviews and Meta-Analysis Protocols checklist, with all screening steps and quality appraisal conducted by two independent reviewers.

⇒ The peer-reviewed, comprehensive search strategy will ensure the inclusion of a wide range of relevant studies, reducing the risk of selection bias.

⇒ High levels of heterogeneity across studies due to differences in populations, condition measurements and study designs may limit the possibility of conducting a meta-analysis.

⇒ Qualitative studies will be excluded therefore limiting an in-depth exploration of the perspectives of adults in Ghana on social determinants associated with their mental health, which may have provided important information to inform future intervention development.

## INTRODUCTION

Common mental disorders, such as anxiety and depression, are among the most prevalent and rising health problems affecting adults worldwide.[1 2] Anxiety and depression are leading causes of health-related burden globally, with depression being the second and anxiety being the eighth leading cause of years lived with disability (YLDs) out of 369 diseases and injuries in 204 countries and territories.[1] The consequences of anxiety and depression can be severe; resulting in impaired social functioning, increased mortality rates, low productivity, and reduced quality of life.[3] Research suggests that anxiety and depression are associated with an increased risk of developing complications or worsening a range of chronic physical

conditions including cancer, diabetes, heart disease, and HIV.[4–6] The economic burden of anxiety and depression is also substantial with costs stemming from healthcare service use, lost productivity, and impact on families, caregivers and wider society.[7 8] The burden and impact of anxiety and depression on individuals and societies underscores the critical need to prioritise adult mental health and well-being globally.

The majority of the global burden of mental disorders is located in low-income and middle-income countries (LMICs),[9] where 75% of individuals experiencing common mental disorders do not receive appropriate mental healthcare.[10] This treatment gap is particularly pronounced in Africa owing to the limited availability of mental health resources, coupled with healthcare systems that are inadequately equipped to meet the needs of individuals seeking care.[11 12] A previous scoping review conducted in 12 African countries found a lifetime prevalence rate of anxiety ranging from 5.7% to 15.8% and depression ranging from 3.3% to 9.8%.[13] Within Africa, the prevalence of mental disorders including anxiety and depression is expected to increase by 130% between 2010 and 2050, potentially leading to 45 million YLDs in sub-Saharan Africa (SSA),[14] with major depression predicted to be the largest contributor to disease burden in SSA.[14] Therefore, researchers, global health entities, and advocacy groups are urging actions to address the growing mental health crisis in SSA,[15] advocating for integrating mental health into primary care, establishing national policies, training paraprofessionals, expanding community healthcare, and collaborating with traditional healers.[14 16] To further inform these actions,[15] there is a need to better understand the determinants of mental health that may contribute to the development and exacerbation of mental disorders in SSA, particularly in resource-constrained countries in the region.

Determinants of mental health among adults are multifaceted, and can be broadly categorised into biological (eg, genetics, brain chemistry and hormonal imbalances), psychological (eg, cognitive and interpersonal factors), and social (eg, ethnicity, food security and neighbourhood deprivation).[17] Social determinants can also exacerbate biological and psychological vulnerabilities via mechanisms such as gene–environment interactions, epigenetic modifications, and maladaptive coping.[18–20] Globally, social determinants are recognised for their role in either providing protection or increasing the risk of adult mental disorders such as anxiety and depression, while also contributing to health disparities.[21–23] Within LMICs, including those in Africa, social determinants including diminished social capital, environmental events, food insecurity, forced migration, income inequality, violence, low education, poor housing, poverty and unemployment,[24–27] may increase vulnerability to experiencing mental health difficulties.[15] Ethnicity as a social determinant also requires additional consideration given experiences of discrimination and exclusion, as a result of ethnicity, impact mental health.[22] Furthermore, while ethnicity

has been established as a social determinant of mental health, the interactions of mechanisms such as norms, differences in cultural interpretations of symptoms, and practices can also impact mental health.[21 22] Recognising the over-arching importance of social determinants, the WHO has urged focus on reshaping economic, physical, and social factors to improve mental health and reduce inequalities by accelerating the implementation of the Comprehensive Mental Health Action Plan 2013–2030.[28] Importantly, the majority of social determinants such as poor living arrangements and low or worsening socioeconomic status are modifiable,[29] and may be improved by implementing appropriate governmental policies and adopting a multisectoral approach to provide comprehensive mental health and social care services.[7 30] In SSA, there is a growing consensus that mental health policies and services should be tailored to the contextual realities of each country in the region, with evidence emphasising the importance of documenting the social determinants specific to each SSA country where mental disorders are experienced.[15 31]

The focus of the present review is Ghana, an LMIC in SSA with a population of approximately 31 million, with an estimated 13% of adults experiencing mental disorders including anxiety and depression.[32] Despite this, only 2% receive treatment.[33] Chronic physical conditions like cancer, diabetes and HIV are commonly linked to mental disorders among adults in Ghana[34 35] and the burden of anxiety and depression in Ghana is further worsened by inadequate mental health resources and healthcare disparities.[35] Social determinants such as crime, food insecurity, poor sanitation, poverty, and unemployment may also contribute to elevated rates of mental disorders in adults in Ghana.[35–38] To the best of our knowledge, there has been no systematic review undertaken to consolidate existing research, examine the prevalence rates of anxiety and depression among adults and explore key social determinants specific to the Ghanaian context. By examining the prevalence of anxiety and depression among adults in Ghana and exploring social determinants potentially associated with anxiety and depression, this review will provide insights into the magnitude of the problem in Ghana, and social determinants influencing their occurrence to inform evidence-based policies and interventions aimed at enhancing mental well-being and overall health of adults in Ghana. The review's findings will also be valuable for countries experiencing similar social determinants as the Ghanaian population.

### Research objectives
This systematic review seeks to (1) examine the prevalence of anxiety and depression among adults in Ghana and (2) explore social determinants potentially associated with anxiety and depression.

## METHODS

This review protocol adheres to the Preferred Reporting Items for Systematic reviews and Meta-Analysis Protocols (PRISMA-P) checklist[39] (online supplemental appendix 1). The Joanna Briggs Institute (JBI) methodology for systematic reviews on prevalence was used to guide protocol development.[40] The protocol has been registered in the International Prospective Register of Systematic Reviews. The study is scheduled to commence on 25 September 2023, with an anticipated completion date of 1 September 2024. Any protocol amendments will be recorded in PROSPERO.

### Eligibility criteria

Eligibility criteria was developed and defined in accordance with the CoCoPop (Condition, Context, and Population) and type of studies framework.[41]

### Population

Adults (aged≥18 years) living in Ghana will be eligible for inclusion. Studies solely focused on children/adolescents will be excluded. Studies focusing on both adults and adolescents will be excluded if they do not present data separately for adults and adolescents or data cannot be obtained via correspondence with study authors. Studies conducted with specific subpopulations, including individuals with known psychiatric conditions, prisoners, individuals accused of witchcraft and women with fertility or gynaecological disorders, will be excluded due to their unique circumstances, which predispose them to a higher risk or potentially elevate the likelihood of experiencing mental health problems compared with the general population.

### Condition

Conditions eligible for inclusion include anxiety and/or depression assessed using a structured diagnostic clinical interview in accordance with the International Classification of Diseases and Related Health Problems (ICD-10 or ICD-11), or the Diagnostic and Statistical Manual of Mental disorders (DSM), third, fourth or fifth edition (DSM-III, DSM-IV or DSM-V) such as the Structured Clinical Interview for (SCID), the Mini-International Neuropsychiatric Interview (MINI) or the Composite International Diagnostic Interview (CIDI). Studies using a self-report, clinician or proxy administered screening tool for anxiety (eg, Beck Anxiety Inventory), depression (eg, Beck Depression Inventory), or psychological distress (eg, Kessler Psychological Distress Scale) will also be eligible for inclusion. Studies reporting point (current), period (timepoint) or lifetime estimates of the prevalence of anxiety, depression, and psychological distress will be included.[42] To ensure the reliability and quality of assessment methods, only studies using instruments validated for use in adult populations will be included. Validity will be assessed based on evidence provided in the validation paper(s) of the measurement instrument and/or evidence of psychometric properties such as construct validity, content validity, criterion validity, and reliability measures including internal consistency, test–retest reliability, and inter-rater reliability. Studies using instruments validated for use in adult samples in Ghana will also be eligible for inclusion. Studies focused on other mental disorders (eg, psychotic disorders and bipolar affective disorders), substance use (eg, alcohol dependence), and neurological disorders (eg, multiple sclerosis) will be excluded. Studies will be excluded if the prevalence of anxiety, depression, and psychological distress cannot be calculated, for example, when reported solely as mean score or due to insufficient data.

### Context

This review will include studies conducted in Ghana with adults sampled from the community or clinical settings (eg, primary healthcare facilities and hospitals). Studies conducted in regions that encompass Ghana (eg, West Africa and SSA) will be considered eligible if data on participants living in Ghana can be extracted from the publication or obtained via correspondence with study authors. Studies conducted immediately after conflict (ie, less than four months after the official end date of the conflict),[43] humanitarian crises, or natural disaster will be excluded as we seek to understand the general prevalence of anxiety and depression.[44]

### Types of studies

Primary quantitative studies, with observational study designs including longitudinal cohort studies (baseline data only), case–control and cross-sectional studies reporting the prevalence of anxiety, depression and/or psychological distress among adults in Ghana will be included. Mixed-method studies will be eligible for inclusion only if data from the quantitative component can be clearly extracted. In the case of studies conducted on the same cohort of individuals at the same or different points in time, or where samples overlap, only the study with the largest sample and findings related to the aims of this review will be included to ensure that duplicate data are not included. Studies such as case reports, commentaries, conference proceedings, editorials, letters, opinion papers, qualitative studies, reviews, and theses/dissertations will be excluded.

### Information sources

Searches will be conducted in accordance with PRISMA 2020 guidelines.[45] Electronic database searches will be carried out in African Index Medicus (AIM), African Journals Online (AJOL), Cumulative Index to Nursing and Allied Health Literature (CINAHL), Excerpta Medica Database (Embase), Ghana Medical Journal (GMJ), Health Sciences Investigation (HIS), MEDLINE (PubMed), PsycINFO, and SCOPUS. Electronic databases will be searched from inception up to 25 September 2023, and updated study searches will be conducted within 3 months to submitting the results manuscript. Reference lists of included studies will be manually checked

and forward citation checks of included studies will be performed. Reference lists of relevant systematic reviews conducted in SSA, including Ghana will also be manually checked. Grey literature will be searched in Agency for Healthcare Research and Quality (AHRQ), Google Scholar, Health Systems Trust, Open Grey (http://www.opengrey.eu/), and the WHO Institutional Repository for Information Sharing (WHO IRIS). Researchers and non-governmental organisations working in the area of adult mental health in Ghana will be contacted to identify unpublished or ongoing studies.

### Search strategy

The search strategy has been developed in collaboration with Mattias Axén, a librarian at Uppsala University Library and was reviewed by Alkistis Skalkidou and Lene Lindberg, following the PRESS Peer-Review guidelines[46] (online supplemental appendix 2). The search was constructed using terms related to (1) mental disorders and (2) Ghana (see online supplemental appendix 3). Electronic databases will be searched using Medical Subject Headings when possible and free text words in title and abstract word searches. No date restriction will be imposed and only studies published in English and Ghanaian languages (eg, Fante, Ga, and Twi) will be considered for inclusion.

### Study selection

Studies retrieved from searches will be uploaded into Endnote V.20 with duplicates identified and removed. Two reviewers will independently screen titles and abstracts in Rayyan,[47] followed by full paper checks of potentially eligible studies. Studies not meeting the eligibility criteria will be excluded. Overall reasons for exclusion will be documented and reported using the PRISMA flow chart and detailed reasons for exclusion, in accordance with the eligibility criteria, will be presented in a table. If study data/information required to determine eligibility is missing, authors will be contacted at most twice over a 1-month period via email for additional information. If the authors do not respond, the study will be excluded. Any disagreement between reviewers will be resolved by discussion and/or involvement of a third reviewer.

### Data extraction

Data from included studies (see online supplemental appendix 4) will be extracted by one reviewer independently into a standardised Microsoft Excel data extraction form, and crosschecked by a second reviewer for accuracy. The following data will be extracted:

1. Study identification features: study ID/record number, study title and aim, first author name, year of publication, and publication type or data source (eg, journal or report).
2. Study characteristics: characteristics of study population (age, gender, socioeconomic status, and chronic physical condition reported), sample size, sampling methods (eg, convenience sampling and random sampling), study setting including geographical region(s) in Ghana where study was conducted, location (rural, urban, and mixed) and recruitment setting (community or clinical settings), study design (eg, case–control, cross-sectional, and longitudinal cohort studies), time period of data collection, type of mental health condition (anxiety, depression, and psychological distress), structured diagnostic clinical interview (yes/no), name of diagnostic clinical interview, screening tool (yes/no), name of screening tool and cut-off scores, evidence of validity of the measurement instrument, as well as evidence of ethical approval.
3. Results summary: binary prevalence data of anxiety, depression and psychological distress (ie, number of cases/ the total sample size, n/N) and percentage with 95% CIs, prevalence type (current, period, or lifetime), and summary of social determinants classified into demographic, economic, environmental, neighbourhood, social, and cultural domains.

### Quality assessment

The methodological quality of all included studies will be independently assessed by two reviewers using the standardised JBI Critical Appraisal Tool for Prevalence Studies.[48] The JBI Critical Appraisal Tool for Prevalence Studies comprises nine items including appropriateness of sampling frame and sampling technique, adequacy of sample size, coverage of identified sample, description of study subjects and setting, validity of condition identification methods, standard and reliable measurement of condition, statistical analysis and adequacy of response rate to assess the methodological quality and/or risk of bias of sampling, analysis and measurements in primary studies. The total score ranges from 1 to 9 for individual studies,[49] with the total number of 'yes' scores for individual studies averaged to appraise studies as low (≤3 score), moderate (4–6 points) or high quality (≥7 points).[50] Any disagreement between reviewers will be resolved by discussion and/or involvement of a third reviewer.

### Data analysis and synthesis

#### Quantitative data synthesis

If data allow, a meta-analysis with prevalence data from eligible studies will be conducted using Comprehensive Meta-Analysis software.[51] Data on the proportion of adults with anxiety, depression, and psychological distress and their respective sample size will be extracted separately from individual studies to generate pooled estimates with exact binomial test and associated 95% CIs. Data will be transformed to their logits before meta-analysis to stabilise variances.[52] Due to expected heterogeneity in individual studies, the random-effect model will be used to generate pooled prevalence estimates for anxiety, depression, and psychological distress, respectively.[51] Heterogeneity across studies will be estimated using Cochran's Q statistic. The $I^2$ statistics will be used to measure the proportion of total variability due to between-study

heterogeneity. The prediction interval ($T^2$) will be used as an estimate of between-study variance in true effects observed in eligible studies. Sensitivity analyses will be performed to explore the impact of individual studies on the overall prevalence estimate of anxiety and depression. This will be conducted, for example, by removing studies conducted during COVID or studies of lower quality individually from the overall analysis to ascertain if their removal causes any substantial change to overall prevalence estimates. Egger's regression statistic and funnel plots will be used to assess the presence of any publication bias.[53] In the case of significant publication bias, the trim and fill method will be used to identify and correct the asymmetry of the funnel plot to yield a corrected pooled prevalence.

### Subgroup analysis

If data allow, we will explore sources of heterogeneity via subgroup analysis of the moderating effects of the following factors on prevalence:

► Chronic physical conditions (eg, long-term life-threatening conditions such as cancer, diabetes, emphysema, hypertension, HIV/AIDS, ischaemic heart disease and stroke, and chronic conditions such as arthritis, asthma, back problems of any kind, chronic bronchitis, gall bladder diseases, joint pain, osteoporosis, and stomach ulcers).[54]
► Method of mental health assessment (eg, structured clinical interview or screening tool).
► Sample size (eg, <100 vs ≥100).
► Study design (eg, case–control study, cross-sectional studies and longitudinal cohort study).
► Study quality (low, medium, or high-quality studies).
► Prevalence type (point, period, and lifetime).
► Time period of data collection or publication year (eg, <2013 vs ≥2013 or <2020 vs ≥2020).

### Narrative synthesis

Social determinants will be narratively synthesised, with specific distal and/or proximal factors grouped under domains of the social determinants of mental disorders and Sustainable Development Goals (SDGs) framework,[22] outlined as follows:

1. Demographics: proximal factors (eg, age, ethnicity, and gender) and distal factors (eg, community diversity and population density).
2. Economic: proximal factors (eg, assets, debt, and unemployment) and distal factors (eg, economic inequality and recessions).
3. Neighbourhood: proximal factors (eg, safety and security, housing structure, and overcrowding) and distal factors (eg, neighbourhood deprivation and built environment).
4. Environmental events: proximal factors (eg, trauma) and distal factors (climate change, forced migration, war or conflict).
5. Social and cultural: proximal factors (eg, individual social capital, social participation, and education) and distal factors (eg, social stability and community social capital).

### Patient and public involvement

Public contributors were not involved in the development of this protocol and will not be involved in the conduct of the review due to time and resource limitations. We will seek to involve public contributors in cowriting plain language summaries to be provided to non-governmental organisations working in the area of adult mental health in Ghana.

## DISCUSSION

To the best of our knowledge, there is currently no comprehensive review examining the prevalence and social determinants of anxiety and depression among adults in Ghana. This review will extend existing epidemiology literature on mental health[36 55 56] by providing a comprehensive summary of prevalence estimates of anxiety and depression among adults in Ghana, as well as social determinants potentially influencing their occurrence. The review has a number of strengths including a peer-reviewed, comprehensive search strategy ensuring the inclusion of a wide range of relevant studies, and reducing the risk of selection bias. By considering comorbidities, the review recognises that mental health difficulties often co-occur with physical conditions, contributing to a better understanding of the overall health and mental well-being of adults in Ghana. Comorbidity between anxiety and depression is frequent and if data allows, the prevalence of comorbid anxiety and depression will be documented. The review protocol also adheres to quality standards informed by the PRISMA-P checklist,[39] with screening, selection and quality appraisal assessed by two independent reviewers.

While there are several strengths, there are also limitations. Qualitative studies will be excluded, limiting an in-depth exploration of the perspectives of adults in Ghana on social determinants associated with their mental health, which may have provided important information to inform mental health intervention development. High levels of clinical and methodological heterogeneity across studies due to factors such as differences in populations, condition measurements and study design may limit the possibility of conducting a meta-analysis. This situation may warrant the need for narrative synthesis.

Despite these limitations, the significance of this review is highlighted by the widespread global attention on mental health, with initiatives such as the WHO Comprehensive Mental Health Action Plan 2013–2030 and the United Nation's SDGs highlighting the importance of mental healthcare equity, prevention, treatment and promotion.[57] While Ghana is working towards improving mental health via its revised 12 year Mental Health Policy launched in 2021,[35] a comprehensive review examining the prevalence of anxiety and depression among adults in Ghana can facilitate the planning and allocation of

resources for mental healthcare.[35] Addressing the social determinants is also crucial for achieving broader development targets, such as SDG 1 for no poverty, SDG 5 for gender equality and SDG 4 for quality education, among others.[22] Findings may be used to inform future research, mental healthcare planning and the development of culturally responsive interventions aimed at improving the mental well-being and overall health of adults in Ghana.

**Acknowledgements** We thank Alkistis Skalkidou (AS) from the Department of Women's and Children's Health at Uppsala University; Lene Lindberg (LL) from the Department of Public Health Sciences at Karolinska Institute for providing peer review of the search strategy; Mattias Axén (MA), a Librarian at Uppsala University Library, for assisting with the development of the electronic search strategy and Febrina Maharani (FM), master student in Global Health at Uppsala University, for assisting with study selection process.

**Collaborators** N/A.

**Contributors** VA, MD and JW conceptualised the study. VA drafted the proposal. VA, CC and JW designed the study. All authors (VA, MD, EO, SA, CC and JW) assisted with manuscript writing and critical revision of the study design and manuscript. All authors read and approved the final manuscript. JW is the guarantor of the review.

**Funding** This work was supported by U-CARE, which is a strategic research environment funded by the Swedish Research Council (dnr 2009–1093).

**Disclaimer** This funding source had no role in the design of this study and will not have any role during its execution, analyses, interpretation or the data or decision to submit results

**Competing interests** None declared.

**Patient and public involvement** Patients and/or the public were involved in the design, or conduct, or reporting or dissemination plans of this research. Refer to the Patient and public involvement section for further details.

**Patient consent for publication** Not applicable.

**Provenance and peer review** Not commissioned; externally peer reviewed.

**ORCID iDs**
Victoria Awortwe http://orcid.org/0000-0002-9851-1655
Meena Daivadanam http://orcid.org/0000-0002-9532-6059
Samuel Adjorlolo http://orcid.org/0000-0001-9308-6031
Erik MG Olsson http://orcid.org/0000-0002-1591-7407
Chelsea Coumoundouros http://orcid.org/0000-0001-5539-974X
Joanne Woodford http://orcid.org/0000-0001-5062-6798

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
