## [Reviewer comments · BMJ Open]

ARTICLE DETAILS

TITLE (PROVISIONAL)	Prevalence and social determinants of anxiety and depression among adults in Ghana: a systematic review and meta-analysis protocol
AUTHORS	Awortwe, Victoria; Daivadanam, Meena; Adjorlolo, Samuel; Olsson, Erik; Coumoundouros, Chelsea; Woodford, Joanne

VERSION 1 – REVIEW

REVIEWER	Grummitt, Lucinda The University of Sydney
REVIEW RETURNED	12-Jan-2024

GENERAL COMMENTS	This was a well-written protocol describing a planned systematic review and possible meta-analysis (depending on heterogeneity) to determine the prevalence of anxiety and depression among adults in Ghana, and the social determinants of depression and anxiety in Ghanaian adults. It is an important topic, and the authors should be commended on a very well-written introduction, which nicely provides a summary of the relevant background literature and rationale for the present study in a concise manner. The manuscript could be improved through attention to the following: Introduction Page 5, line 7: The example of “ethnicity” as a social determinant of mental health stood out to me. Perhaps “culture” is more appropriate here, or if the authors are referring to racism or discrimination as social determinants, those terms should replace ethnicity. Methods Condition: Are the authors proposing to only include studies that use the latest versions of ICD and DSM in their assessment of mental health? This appears to be the case from the text, however this would seriously restrict the number of eligible studies and thus I encourage the authors to extend the inclusion criteria to at least DSM-IV. Page 7, line 17: It would be important to include precise criteria regarding validated instruments. For example, can instruments be validated with any sample, adult samples only, adult samples from LMICs only? Page 7, line 34: Please clarify what timeframe constitutes “immediately after conflict” i.e., within one week of conflict ending, one month, 6-months etc.
---

	Page 8, line 3: Include the planned date of study searches e.g., databases will be searched from inception through to the date of study searches (anticipated month/year).
--	--

VERSION 1 – AUTHOR RESPONSE

Reviewer 1

Thank you for your very helpful comments. Please find how we have addressed each comment below:

This was a well-written protocol describing a planned systematic review and possible meta-analysis (depending on heterogeneity) to determine the prevalence of anxiety and depression among adults in Ghana, and the social determinants of depression and anxiety in Ghanaian adults. It is an important topic, and the authors should be commended on a very well-written introduction, which nicely provides a summary of the relevant background literature and rationale for the present study in a concise manner. The manuscript could be improved through attention to the following:

Introduction

Page 5, line 7: The example of “ethnicity” as a social determinant of mental health stood out to me. Perhaps “culture” is more appropriate here, or if the authors are referring to racism or discrimination as social determinants, those terms should replace ethnicity.

Thank you for your comment. We would like to retain the term “ethnicity” as it is a specific factor in the framework that we are using. We think that ethnicity is a better term to use than discrimination or racism when discussing the social determinants of mental health because it encompasses a broader range of factors beyond discrimination or racism. While discrimination and racism are significant components of how ethnicity can impact mental health, ethnicity also includes, for example, cultural background, differential cultural interpretation of symptoms, exclusion, genetic background, all of which can influence mental health. Using the term ethnicity acknowledges the complexity of these factors and their interplay with mental health outcomes. Additionally, ethnicity allows a more nuanced understanding of how various aspects of identity intersect and contribute to mental health disparities. Further, the term “racism” is not used in the literature from Ghana largely because racism is not mentioned or even discussed in Ghana. Rather, discussions often tend to centre on ethnic background of the population. With several ethnic groups having their unique cultural practices, it is important to understand how ethnicity impact mental health.

Methods

Condition: Are the authors proposing to only include studies that use the latest versions of ICD and DSM in their assessment of mental health? This appears to be the case from the text, however this would seriously restrict the number of eligible studies and thus I encourage the authors to extend the inclusion criteria to at least DSM-IV.

Thank you for this comment. On page 7, we have now extended the inclusion to the third and fourth version of DSM as follows:

The Diagnostic and Statistical Manual of Mental disorders (DSM), third, fourth or fifth edition (DSM-III, DSM-IV or DSM-V).

Page 7, line 17: It would be important to include precise criteria regarding validated instruments. For example, can instruments be validated with any sample, adult samples only, adult samples from LMICs only?

Thank you for this comment. We have added some additional detail on the precise criteria regarding validated instruments on Page 7:

only studies using instruments validated for use in adult populations will be included. Validity will be assessed based on evidence provided in the validation paper(s) of the measurement instrument and/or evidence of psychometric properties such as construct validity, content validity, criterion validity, and reliability measures including internal consistency, test-retest reliability, and inter-rater reliability. Studies using instruments validated for use in adult samples in Ghana will also be eligible for inclusion.

Page 7, line 34: Please clarify what timeframe constitutes “immediately after conflict” i.e., within one week of conflict ending, one month, 6-months etc.

We have now provided additional clarification regarding what constitutes “immediately after conflict”. Specifically, we now define the timeframe as less than four months after the official end date of the conflict. This timeframe aligns with the approach used by Lim et al. (2022) in their study titled “Prevalence of depression, anxiety and post-traumatic stress in war-and conflict-afflicted areas: A meta-analysis.” In their study, post-conflict study was considered eligible for inclusion if it was collected at least four months after the official end date of the conflict. Please see the context section on page 7:

Studies conducted immediately after conflict (i.e., less than four months after the official end date of the conflict),⁴³

Page 8, line 3: Include the planned date of study searches e.g., databases will be searched from inception through to the date of study searches (anticipated month/year).

Thank you for your comment. We have now added the planned date of study searches. Please see the information sources section on page 8:

Electronic databases will be searched from inception up to September 25, 2023, and updated study searches will be conducted within 3 months to submitting the manuscript results.

Point-by-point Amendment

In order to further improve the methodological quality of the review, we would like to make some minor changes to the protocol which we have presented after the response to the editor and reviewer and highlighted in green in the manuscript.

Research objectives:

On page 6, we have revised the objectives to include the combined symptom measure of anxiety and depression, which is psychological distress:

This systematic review seeks to: (1) examine the prevalence of anxiety, depression and psychological distress among adults in Ghana (2) explore the social determinants potentially associated with anxiety, depression, and psychological distress.

Eligibility criteria in the method section:

We have added some additional details regarding the exclusion criteria for population. Please see the last three sentences on page 6, as well as the first two sentences of page 7:

Studies conducted with specific sub-populations including individuals with known psychiatric conditions, prisoners, individuals accused of witchcraft, and women with

fertility or gynaecological disorders, will be excluded due to their unique circumstances, which predispose them to a higher risk or potentially elevate the likelihood of experiencing mental health problems compared to the general population.

We have added some additional details regarding the exclusion criteria for condition. Please see page 7:

Studies will be excluded if the prevalence of anxiety, depression and psychological distress cannot be calculated, for instance, when reported solely as mean score or due to insufficient data.

We have added some additional details regarding the inclusion and exclusion criteria of type of studies. Please see page 8:

Primary quantitative studies, with observational study designs including longitudinal cohort studies (baseline data only), case-control, and cross-sectional studies reporting the prevalence of anxiety, depression and/or psychological distress among adults in Ghana will be included. Mixed method studies will be eligible for inclusion only if data from the quantitative component can be clearly extracted. In the case of studies conducted on the same cohort of individuals at the same or different points in time; or where samples overlap, only the study with the largest sample and findings related to the aims of this review will be included to ensure duplicate data is not included. Studies such as case reports, commentaries, conference proceedings, editorials, letters, opinion papers, qualitative studies, reviews, theses/dissertations will be excluded.

Data extraction

We have now included the extraction of data on cut-off scores, evidence of validity of the measurement instrument, and the evidence of ethical approval. Additionally, in the result summary, we have removed redundant words. Please see the last two lines of the study characteristics on page 10:

evidence of validity of the measurement instrument, as well as evidence of ethical approval.

And the first three lines of the result summary on page 10, where redundant words have been removed:

binary prevalence data of anxiety, depression and psychological distress (i.e. number of cases/ the total sample size, n/N) and percentage with 95% confidence intervals, prevalence type (current, period or life time) ...

Quantitative data synthesis

We have added that sensitivity analyses will also be conducted by removing studies conducted during Covid to ascertain if their removal causes any substantial change to overall prevalence estimates.

Page 11:

Sensitivity analyses will be performed to explore the impact of individual studies on the overall prevalence estimate of anxiety and depression. This will be conducted for example, by removing, studies conducted during COVID and/or studies of lower quality individually from the overall analysis to ascertain if their removal causes any substantial change to overall prevalence estimates.

We have added some additional details clarifying what chronic physical conditions constitute and a reference:

Page 11, in the sub-group analysis section:

Chronic physical conditions include long-term life-threatening conditions such as Cancer, Diabetes, Emphysema, Hypertension, HIV/AIDS, Ischemic Heart Disease, and Stroke, and chronic manageable conditions like Arthritis, Asthma, Back Problems of any kind, Chronic Bronchitis, Gall Bladder Diseases, Joint Pain, and Stomach Ulcers. However, mental health conditions listed in the compilation like Alzheimer’s Diseases and other Dementias, anxiety, Attention Deficit Hyperactivity Disorder, (ADHD)/Attention Deficit Disorder (ADD) and depression⁵⁴ were excluded due to our study’s focus on anxiety and depression, along with their combined symptom measure, while excluding other mental health conditions.

Discussion

We have included a sentence in the discussion section addressing potential reporting of comorbid anxiety and depression data:

Page 13:

Comorbidity between anxiety and depression is frequent and if data permits, the prevalence of comorbid anxiety and depression will be documented

VERSION 2 – REVIEW

REVIEWER	Grummitt, Lucinda The University of Sydney
REVIEW RETURNED	28-Mar-2024

GENERAL COMMENTS	I thank the authors for considering my first point around the use of the term ethnicity as a social determinant of health. While I understand their points, it is similar to the argument around race and racism i.e., it is racism, and not race, that drives inequities in health and mental health, and it is potentially harmful to talk about race and ethnicity themselves as drivers of health inequity. The authors’ response was clear and placated me on this issue, however, some of this detail should be included in the manuscript to make this nuance clear to readers. For example, mentioning explicitly that ethnicity as a social determinant of health captures / refers to things such as how cultural background influences mental health. The authors have addressed all my other concerns.
--

VERSION 2 – AUTHOR RESPONSE

Dear Lucinda Grummitt,

Thank you for your very helpful comment. Please find how we have addressed the comment below:

I thank the authors for considering my first point around the use of the term ethnicity as a social determinant of health. While I understand their points, it is similar to the argument around race and racism i.e., it is racism, and not race, that drives inequities in health and mental health, and it is potentially harmful to talk about race and ethnicity themselves as drivers of health inequity.

The authors' response was clear and placated me on this issue, however, some of this detail should be included in the manuscript to make this nuance clear to readers. For example, mentioning explicitly that ethnicity as a social determinant of health captures / refers to things such as how cultural influences mental health. The authors have addressed all my other concerns.

We have included some additional detail in the manuscript regarding ethnicity on Page 5:

Ethnicity as a social determinant also requires additional consideration given experiences of discrimination and exclusion, as a result of ethnicity, impact mental health.[22] Furthermore, while ethnicity has been established as a social determinant of mental health, the interactions of mechanisms such as norms, differences in cultural interpretations of symptoms and practices can also impact mental health.[21, 22]